# Multilevel Analysis of Urban–Rural Variations of Body Weights and Individual-Level Factors among Women of Childbearing Age in Nigeria and South Africa: A Cross-Sectional Survey

**DOI:** 10.3390/ijerph19010125

**Published:** 2021-12-23

**Authors:** Monica Ewomazino Akokuwebe, Erhabor Sunday Idemudia

**Affiliations:** Faculty of Humanities, North-West University, Mafikeng 2745, South Africa; erhabor.idemudia@nwu.ac.za

**Keywords:** body weight, clustering areas, factors, women of childbearing age, Nigeria, South Africa

## Abstract

*Background*: An unhealthy body weight is an adverse effect of malnutrition associated with morbidity among women of childbearing age. While there is increasing attention being paid to the body weights of children and adolescents in Nigeria and South Africa, a major surge of unhealthy body weight in women has received less attention in both countries despite its predominance. The purpose of this study was to explore the prevalence of body weights (underweight, normal, overweight, and obese) and individual-level factors among women of childbearing age by urban–rural variations in Nigeria and South Africa. *Methods*: This study used the 2018 Nigeria Demographic Health Survey data (*n* = 41,821) and 2016 South Africa Demographic Health Survey (*n* = 8514). Bivariate, multilevel, and intracluster correlation coefficient analyses were used to determine individual-level factors associated with body weights across urban–rural variations. *Results*: The prevalence of being overweight or obese among women was 28.2% and 44.9%, respectively, in South Africa and 20.2% and 11.4% in Nigeria. A majority, 6.8%, of underweight women were rural residents in Nigeria compared to 0.8% in South Africa. The odds of being underweight were higher among women in Nigeria who were unemployed, with regional differences and according to breastfeeding status, while higher odds of being underweight were found among women from poorer households, with differences between provinces and according to cigarette smoking status in South Africa. On the other hand, significant odds of being overweight or obese among women in both Nigeria and South Africa were associated with increasing age, higher education, higher wealth index, weight above average, and traditional/modern contraceptive use. Unhealthy body weights were higher among women in clustering areas in Nigeria who were underweight (intracluster correlation coefficient (ICC = 0.0127), overweight (ICC = 0.0289), and obese (ICC = 0.1040). Similarly, women of childbearing age in clustering areas in South Africa had a lower risk of experiencing underweight (ICC = 0.0102), overweight (ICC = 0.0127), and obesity (ICC = 0.0819). *Conclusions*: These findings offer a deeper understanding of the close connection between body weights variations and individual factors. Addressing unhealthy body weights among women of childbearing age in Nigeria and South Africa is important in preventing disease burdens associated with body weights in promoting Sustainable Development Goal 3. Strategies for developing preventive sensitization interventions are imperative to extend the perspectives of the clustering effect of body weights on a country level when establishing social and behavioral modifications for body weight concerns in both countries.

## 1. Introduction

Worldwide, malnutrition is a public health, social, and economic problem, imposing high human capital costs directly and indirectly on individuals, families, and nations. Several studies have estimated that all forms of malnutrition may perhaps cost society up to 3.5 trillion USD per year, with overweight and obesity alone having an estimated cost of about 500 billion USD per year [1,2]. Thus, the public health concerns associated with growing prevalence of childhood deaths and impending adult disability, with diet-related noncommunicable diseases (NCDs), have imposed enormous economic and human capital costs [3,4]. Thus, the Sustainable Development Goals (SDGs), under the auspices of the World Health Organization and the 2030 Agenda for Sustainable Development, have committed to reducing one-third of the burden of NCDs through prevention and treatment (SDG target 3.4). Several studies have cited a major surge in nutrition transition cycle, resulting in a burden of all forms of malnutrition, especially in developing countries such as those in Africa [5,6].

Moreover, malnutrition (leading to underweight, overweight, and obesity) has posed serious health risks with adverse implications for population health. In 2014, about 462 million adults globally were underweight, and 1.9 billion were either overweight or obese [7]. In 2016, approximately 41 million children under the age of 5 years were overweight or obese, whereas 155 million were persistently malnourished [8]. Accordingly, being overweight or obese is one of the leading risk factors for several NCDs and other chronic medical conditions. Furthermore, women with non-normal body weights are prone to various adverse diseases that are associated with detrimental health conditions and increased risk of early mortality. Many studies have revealed that women are more susceptible to malnutrition and its health risks owing to the interplay between food consumed and their genetic makeup [6,9].

Although the prevalence of underweight is decreasing in high-income countries, the upsurge of underweight, overweight, and obesity across African countries such as Nigeria and South Africa has become a public health concern. In Nigeria, studies have reported epidemiological and demographical changes, as well as nutritional transitions, which are driven by urbanization and unhealthy lifestyles, as the leading contributors to underweight and overweight/obesity [10,11]. According to the Nigeria Demographic Health Survey (NDHS) [11], there is an alarming increase in the prevalence of underweight (12%) and overweight/obesity (28%) among women of childbearing age (15−49 years). Moreover, the NDHS trend analysis has shown that the percentage of thin women aged 15–49 years has remained constant over the past 10 years at 12%, whereas the percentage of those who are overweight/obese has increased from 22% in 2008 to 28% in 2018 [11].

In South Africa, malnutrition has become an emergent public health problem, as the country is going through an epidemiological health transition revealing prevailing chronic malnutrition [12]. Factors such as increased adoption of more westernized diets and the rise in sedentary behavior, owing to modernization, improved transport systems, and easy convenient access to unhealthy fast foods, are associated with over-weight/obesity and its related-health problems in South Africa [6,13]. Studies have documented that the body mass index (BMI) cutoffs for women and men are 29.2 and 23.6 respectively, whereby 68% of women are overweight/obese while 3% are underweight, as two-thirds of women (59%) have a BMI in the standard range [14]. According to the South Africa Demographic Health Survey (SADHS), the trend analysis indicated that the mean BMI among women aged 15 and older has increased from 27.3 in 1998 to 29.2 in 2016, and the prevalence of overweight/obesity among women of childbearing age rose from 56% to 68%, with a decreased prevalence of underweight from 6% to 3% [12]. In Nigeria and South Africa, underweight prevalence is declining to an extent, in comparison to overweight/obesity, which has a higher prevalence [11,12,15,16]; yet, other countries are still observing an increased prevalence of underweight [17,18]. Hitherto, studies on malnutrition conducted in Nigeria and South Africa have shown that overweight or obese women of childbearing age were more likely to be older, educated, married, in the highest wealth quintile, and residing in urban areas [11,12]. By contrast, women who were never married, resided in poor households, and had lower education attainment were more likely to be underweight in both countries [11,12].

Although useful, these studies have a number of limitations. First, the earlier studies explored population-based subgroups (such as children, adolescents, the elderly, and males) using other national representative datasets and primary-based community surveys, which may not clearly give the true picture of the socioeconomic, demographic, and health status of both countries. Second, the previous studies did not consider the comparative analysis of both countries, as they have indispensable social and demographic dynamics. Moreover, earlier studies did not consider the most suitable method (comprising the categorized nature of data) in handling effectively larger datasets for Nigeria and smaller datasets for South Africa. In contrast, our study uses a multilevel model analysis (linear mixed-effect model) involving a level two regression equation and random intercepts model to weigh the intracluster correlation coefficient (ICC) and a binary logistic regression for bivariate analysis, an important statistical methodology that has not been employed in previous studies conducted in Nigeria and South Africa for comparative analysis according to urban–rural variations.

Understanding the factors associated with underweight and overweight/obesity among women of childbearing age in Nigeria and South Africa would be useful for relevant health and non-health experts to implement evidence-based and appropriate interventions to address all forms of malnutrition. This context-specific information is also important to national and international stakeholders, given the contemporary obligation to accomplish SDG 3 to end all forms of malnutrition, as well as the Global Action Plan for the Prevention and Control of NCDs target 9 to reduce the overweight/obesity burden in both countries [19,20,21,22]. However, there is a dearth of research on the influence of urban–rural residence on women’s body weight in Nigeria and South Africa. Moreover, nationally representative studies on the body weight of women of childbearing age (15–49 years) and associated factors, using a multi-level model and ICC involving a cross-sectional survey and comparative analysis stratified by urban–rural variations in Nigeria and South Africa, are lacking.

We undertook a cross-sectional and descriptive study to explore the prevalence of body weight (underweight, normal, overweight, and obese) and its associated factors among women of childbearing age in Nigeria and South Africa stratified by urban–rural variations. The specific objectives based on urban–rural differences in Nigeria and South Africa were to determine the prevalence of body weights by urban–rural variations, examine the predictive influence of the body weight on the associated factors, and appraise the mediating effects of the levels of body weight on the predictive influence of urban–rural variations on its associated factors among women of childbearing age in Nigeria and South Africa.

## 2. Materials and Methods

### 2.1. Study Setting

#### 2.1.1. Nigeria

Nigeria is the most populous country in Africa with an estimated population of 206 million. Although often pointed out as the “Giant Africa”, owing to its enormous population and economy, it is a multi-national state populated by more than 250 ethnic groups. Well-known with an extensive diversity of cultures, the three major ethnic groups—Hausa-Fulani in the North, Yoruba in the West, and Igbo in the East—comprise over 60% of the total population, and the country is home to Christian, Muslim, and indigenous religions [11]. Administratively, Nigeria is divided into states and 774 local government areas, defined by urban and rural areas. Presently, Nigeria is plagued with Boko Haram conflicts, poverty, malnutrition, diseases, and the burden of youth unemployment. In 2018, the NDHS report on maternal height and body mass index (BMI) showed a prevalence of women aged 15–49 years who were overweight/obese (28%), which is on the rise, compared with women who are underweight (12%) [11].

#### 2.1.2. South Africa

South Africa, officially called the Republic of South Africa (RSA), has over an estimated 59 million people in the southernmost part of Africa, covering an area of 1,221,037 square kilometers of land. South Africa has three capital cities: executive Pretoria, judicial Bloemfontein, and legislative Cape Town, with Johannesburg as the largest city [12]. South Africa is very racially diverse, where about 80% of South Africans are of Black African ancestry, with the remainder divided among other ancestry groups: European (White South Africans), Asian (Indian South Africans), and multi-racial (colored South Africans). South Africa is a multi-ethnic society with nine provinces, encompassing a wide variety of cultures, languages, and religions [12]. Even though South Africa is the most westernized country on the African continent, with a mixed economy and a relatively high gross domestic product (GDP) per capita, yet poverty and inequality remain widespread, with a major youth unemployment problem, as approximately one-quarter of the population is unemployed and living on less than 1.25 USD per day [23]. Furthermore, South African society continues to face steep challenges such as rising crime rates, ethnic tensions, great disparities in housing and educational opportunities, and the AIDS epidemic. In 2016, the SADHS documented a high prevalence of overweight/obesity among women aged 15–49 years (68%), with comparable findings in urban (68%) and non-urban (66%) areas [12].

### 2.2. Study Design

The datasets used in this study, the 2016 SADHS and 2018 NDHS [11,12], were combined to maximize the sample size for each study area. In addition to increasing the number of observations, another advantage of combining two different surveys is that it is anticipated that increasing the overall sample size should lead to reduced sampling errors [24]. The 2016 SADHS was the third nationally representative cross-sectional survey in South Africa. The sampling frame of the survey was determined from the list of primary sampling units (PSUs) of the 2011 National Population and Housing Census (NPHC) [12]. It used a two-stage stratified sampling design, where the first stage consisted of 750 PSUs, with 468 in urban areas and 282 in non-urban areas, from a list of residential dwelling units (DUs) generated from the NPHC of South Africa [12]. The second stage of sampling involved a systematic selection of 20 DUs per residential dwelling unit. From the total of 15,292 surveyed households, one in every three households was randomly selected for anthropometric data, and samples were collected from all nine provinces of South Africa. The numbers of urban and non-urban women interviewed in the cross-sectional survey were 4805 and 3709, respectively, giving a total of 8514 women, yielding a response rate of 86% [12]. The survey collected information on various demographic, socio-economic, and health indicators, including individual characteristics and adult nutrition.

The 2018 NDHS is the sixth nationally representative cross-sectional survey in Nigeria. The sampling frame of the survey was determined from the list of enumeration areas (EAs) of the 2011 National Population and Housing Census (NPHC) of the Federal Republic of Nigeria [11]. The study used a two-stage stratified sampling design, where the first stage consisted of 1400 EAs, with 580 in urban areas and 820 in rural areas being selected, with a probability proportional to the EA size generated from the NPHC of Nigeria [11]. The second stage of sampling involved an equal probability systematic sampling and selection of 30 households per EA in the household list [11]. From the total of 41,668 surveyed households, one in every three households was randomly selected for anthropometry measurements. Samples were drawn and collected from six zones of Nigeria, and the numbers of urban and rural women interviewed in the cross-sectional survey were 19,163 and 22,658 respectively, giving a total of 41,821 women, yielding a response rate of 99% [11]. The survey collected information on various demographic, socioeconomic, and health indicators, including individual characteristics, nutrition of children and women, and women’s nutritional status.

### 2.3. Variables Used in the Study

#### 2.3.1. Outcome of Interest

The body weight derived from the body mass index (BMI) was the dependent variable. It was calculated by dividing the body weight in kilograms by height squared (m^2^). BMI was categorized into four categories: underweight (BMI < 18 kg/m^2^), normal weight (18 kg/m^2^ ≤ BMI < 25 kg/m^2^), overweight (25 kg/m^2^ ≤ BMI < 30 kg/m^2^), and obese (BMI ≥ 30 kg/m^2^), according to the World Health Organization’s (WHO) recommendation [25]. Anthropometric measurements on height and weight were recorded for all women aged 15^+^ years during home visits by trained field researchers using procedures standardized in survey settings. In South Africa, the women’s weights were measured using Seca 213 portable stadiometers, and height was measured in meters using an adjustable Seca 201 measuring tape [12]. In Nigeria, women’s weight measurements were also taken using Seca scales with a digital display, model number SECA 878U, and height was measured in meters using a Shorr Board^®^ measuring board [11]. Thus, each measurement tool was calibrated to maintain accuracy with precision to the nearest one-tenth.

#### 2.3.2. Independent Variables

The present study included individual-level factors such as the demographic, socio-economic, and geographical factors as independent variables to explore the body weights of women by urban–rural residence. Therefore, the major independent variables for the study were residence, education, employment status, wealth index, marital status, geographical zone, province, height, and weight. Other independent variables used in the study were identified from previous studies that established their relationship with body weight [26,27,28,29,30,31,32]. These variables included contraceptive method, breastfeeding, living with partner, currently working, and cigarette smoking.

### 2.4. Measurement of Independent Variables

The individual-level variables were established on the basis of accepted genetically related importance, data structure, and published studies. The below explanatory variables for Nigeria and South Africa were included in the multi-level analysis model of this study. The independent variables used in this study are described in Table 1.

### 2.5. Statistical Analysis

Data analyses were conducted and singled out for the study countries on the basis of the socio-demographic factors/variables featured in the 2018 NDHS and 2016 SADHS. The data were weighted for under-sampling and over-sampling errors as per the survey design using the ‘stata svyset’ command before data analyses. All the analyses were based on women’s body weights by urban–rural differences. Subsequently, the analysis of the data involved univariate analysis of the study population characteristics, as well as the prevalence of women’s body weights and prevalence of BMI categories by urban–rural differences. The descriptive statistics reported the frequencies and percentages to summarize the categorical data extracted from the Nigeria and South Africa DHS, while continuous data were measured in averages (±SD). The women’s body weights were measuring using the BMI classification by adopting the World Health Organization’s (WHO) internationally recognized criteria-based BMI: underweight (BMI < 18 kg/m^2^), normal weight (18 kg/m^2^ ≤ BMI < 25 kg/m^2^), overweight (25 kg/m^2^ ≤ BMI < 30 kg/m^2^), and obese (BMI ≥ 30 kg/m^2^). In addition, bivariate analyses of all the independent variables and women’s body weights were carried out using binary logistic regression that reported the adjusted odds ratio (AOR) in order to ascertain if significant associations existed between women’s body weights and individual-level factors. Lastly, multilevel logistic regression analyses (mixed effect) were used to estimate the effect of all the independent variables on the outcome variable. The regression coefficients of the independent variables were unadjusted (U) and expressed as odds ratio (OR) and 95% confidence interval (CI) for body weight categories (underweight, overweight, and obese) using normal weight as the reference category to evaluate predictors of underweight, overweight, and obese in women by urban–rural variations. A null (random intercept only) model was first fitted to evaluate the intracluster correlation coefficient (ICC) to assess the contributions of residence type to each body weight category. The multi-level analysis adjusts for dependency in data owing to variations in communities surveyed, regions/provinces, and other clustered areas. Therefore, adjusting estimates for this dependency is more accurate than measuring within- and between-cluster variations. Women’s households were nested within the urban–rural residence. Hence, the multilevel model was expressed as
(1)$$In\;\left(\frac{Pij}{1}-Pij\right)=\beta 0K+\beta 1X1j+\beta 2X2j+\ldots +\beta 15X15j+eij,\;\;\;\;\;\;\;\;\;Level\;1,$$
(2)                     β0K=γ00+u0K,                                                                             Level 2,
where $In\;\left(\frac{Pij}{1}-Pij\right)$ is the probability of belonging to one of the body weight categories (BMI), β1X1j+β2X2j+…+β15X15j are the model predictors, β0K is the addictive function, γ00 is the grand mean, and u0K is the level 2 random intercept term.

Similarly, significant factors in the bivariate analysis were included in the multivariate model when the variables perfectly predicted the outcome (multicollinearity), while those without an observation set in the model were dropped. The final model of each body weight featured fewer significant predictors, which was established on urban–rural differences, and a variable with odds ratio greater than 1.00 implied that the variable increased the likelihood of the outcome (body weight) while the opposite was true when the OR was less than 1.00. Moreover, the intracluster correlation coefficient (ICC) was employed to account for the relatedness of clustered data by comparing the variance within clusters with the variance between clusters [33]. The ICC is expressed as
ICC (p)=Sb2/(Sb2+ S2w),
where Sb2 is the between-cluster variance, and S2w is the within-cluster variance in the outcome variable. Only intercept multi-level regression models were used to produce estimates of the ICC [34,35], and the explanatory models were not included in the intercept-only models. Theoretically, as S2w (within-cluster variance) moves toward 0 (zero), ICC gets closer to 1 (one). All statistical analyses were conducted using the Statistical Package for the Social Sciences (SPSS) version 14.0 and Stata version 15.0 (StataCorp, College Station City, TX, USA) with the ‘svy’ command to adjust for sampling weights, clustering effects, and stratification; the 95% CI, with a 5% alpha level of significance, was determined.

### 2.6. Ethical Consideration

The 2018 Nigeria and the 2016 South Africa Demographic Health Surveys can be downloaded from the website and are free to use by researchers for further analysis. In order to access the data from the Demographic Health Survey (DHS) MEASURE, a written request was submitted to the DHS MACRO and electronic permission was granted to use the dataset for this study; this was received from the ICF in May 2021. The DHS ensured international ethical standards of confidentiality, anonymity and informed consent, and availability of de-identified DHS datasets.

## 3. Results

### 3.1. Descriptive Results

#### Characteristics of Respondents

Table 2 illustrates the weighted descriptive statistics for individual-level characteristics (socio-demographic, geographical, and behavioral) among women of childbearing age in Nigeria and South Africa. A total weighted sample of 126,538 Nigerian and 14,144 South African women of childbearing age (aged 15–49 years old) was included in the analysis. A greater proportion of the respondents were aged 35–39 years in Nigeria (21.1%) and 30–34 years (19.0%) in South Africa. More than half of the respondents resided in urban areas of South Africa (63.7%) compared to 38.3% of urban respondents in Nigeria. More women in South Africa had secondary education (72.3%) compared to women in Nigeria (23.6%). Regarding employment status, most Nigerian women were self-employed (92.7%) compared to unemployed women (75.3%) in South Africa. A majority of the women in Nigeria (46.4%) and South Africa (44.0%) were mostly found in the poor wealth index. A higher number of Nigerian women (90.0%) were married compared to women in South Africa (39.5%).

Table 2 also revealed that the majority of the women were found in northwest Nigeria (37.7%) and Gauteng province in South Africa (26.0%). Women’s height and weight were classified on the basis of national averages in Nigeria and South Africa. The study findings revealed that approximately 51.7% and 40.8% had height and weight above the average of 1.58 m and 59.7 kg in Nigeria, while about 54.9% and 47.8% of women had height and weight above the average of 1.58 m and 74.1 kg in South Africa. A greater proportion of the respondents in Nigeria had given birth to seven or more children (39.2%) compared to women in South Africa with fewer births (1–3 children) (68.9%). A majority (83.7%) of the women in Nigeria were nonusers of contraceptives compared to women (43.7%) in South Africa, while the majority (56.0%) of South African women used modern contraceptives compared to Nigerian women (12.1%) (Table 2). Similarly, 32.1% and 11.5% of women in Nigeria and South Africa, respectively, were breastfeeding at the time of the survey, while most of the women reported that they are currently living with their partner. A majority of the women in Nigeria reported having long working hours (74.2%) compared to women in South Africa (40.8%), and about 5.3% of women in South Africa engaged in cigarette smoking compared to women in Nigeria (0.2%).

### 3.2. Overall Prevalence of Body Weight among Women of Childbearing Age by Country

The overall prevalence of body weight among women of childbearing age varied by country. The prevalence of obesity was higher among women of childbearing age in South Africa (44.9%) compared to women in Nigeria (11.4%), and a majority of Nigerian women had a normal weight (59.4%) compared to South African women (25.1%). The prevalence of overweight (28.2%) was higher among women in South Africa compared with women in Nigeria (20.2%). Furthermore, 9.0% of Nigerian women were underweight compared to underweight women in South Africa (1.8%) (Figure 1).

### 3.3. Prevalence of Body Weight among Women of Childbearing Age by Urban–Rural Variations in Nigeria and South Africa

Figure 2 and Figure 3 report the prevalence of body weight among women of childbearing age according to urban–rural variations. Overall, around one-fifth of urban women (27.4%, *n* = 1516) and 17.5% (*n* = 970) of rural women were obese in South Africa compared with 6.6% (*n* = 3111) of urban and 4.8% (*n* = 2234) of rural women in Nigeria. The prevalence of overweight was higher in urban (16.9%, *n* = 933) areas compared to rural areas (11.4%, *n* = 630) in South Africa. In Nigeria, rural women (11.2%, *n* = 5240) were more overweight compared to their urban counterparts (9.0%, *n* = 4239). Thus, normal weight prevalence was higher in rural areas in Nigeria (41.3%, *n* = 19,375) and higher among women in urban areas in South Africa (15.1%, *n* = 838). The prevalence of underweight was generally lower in the urban (0.9%, *n* = 52) and rural (0.8%, *n* = 45) areas of South Africa compared to the urban (2.2%, *n* = 1021) and rural (6.9%, *n* = 3195) areas of Nigeria.

### 3.4. Bivariate Analysis of Body Weight and Its Associated Factors

#### 3.4.1. Bivariate Analysis of Women’s Body Weight and Associated Factors in Nigeria

The bivariate results reported an association between women’s body weight (underweight, overweight, and obese, in reference to normal weight) and its associated factors (individual demographic, geographical, and behavioral); the AOR and 95% CI are shown in Table 3. Age, education, and wealth index were significantly associated with overweight and obesity, but showed no significant association with underweight in Nigeria. The findings revealed that increasing age had a significant positive association with overweight and obesity among women of childbearing age in Nigeria. Thus, women aged 40–44 years were more likely to be associated with overweight (OR = 4.20, *p* < 0.01) and obesity (OR = 71.98, *p* < 0.001) compared to women aged 15–19 years, and women aged 45–49 years were more likely to be associated with overweight (OR = 4.99, *p* < 0.01) and obesity (OR = 98.01, *p* < 0.001) compared to women aged 15–19 years (Table 3).

Regarding education, increasing educational status was associated with being overweight (primary (OR = 1.98, *p* < 0.001), secondary (OR = 2.36, *p* < 0.001), and tertiary (OR = 3.43, *p* < 0.001)) and obese (primary (OR = 2.76, *p* < 0.001), secondary (OR = 3.91, *p* < 0.001), and tertiary (OR = 7.30, *p* < 0.001)). Furthermore, for employment status in Nigeria, self-employed respondents were found to be associated with underweight (OR = 1.49, *p* < 0.001), while self-employed (OR = 4.70, *p* < 0.01) and employed (OR = 9.02, *p* < 0.001) respondents were found to be associated with obesity (Table 3). Moreover, increasing wealth index was found to be associated with overweight and obesity, while currently widowed and divorced/separated women were found to be associated with obesity in urban–rural areas. Similarly, women from the North east and North west were found to be associated with underweight, while those from the South south and South west were more likely to be associated with overweight and obesity among urban and rural women (Table 3).

Women with seven or more children were more likely to be associated with underweight, while women with four to six children were more likely to be associated with overweight and obesity. Similar, among urban and rural areas, women who engaged in traditional/modern contraceptive methods were found to be associated with overweight and obesity. Women who were breastfeeding and currently living with their partner at the time of the survey were found to be associated with underweight. However, in urban–rural areas, women who smoked cigarettes had higher odds of being obese (OR = 4.52, *p* < 0.001) (Table 3).

#### 3.4.2. Bivariate Analysis of Women’s Body Weight and Its Associated Factors in South Africa

The associations of underweight, overweight, and obesity with its associated factors (demographic, geographical, and behavioral) of women of childbearing age in rural and urban areas from bivariate analysis showing the AOR and 95% CI are presented in Table 4. Older age, tertiary education, being employed, and the richest wealth index were positively associated with overweight and obesity among women in urban–rural areas in South Africa. Women who were married (OR = 2.52, *p* < 0.001) and divorced/separated (OR = 2.34, *p* < 0.001) had higher odds of being obese than single women, while women who are widowed were found to have higher odds of being both underweight (OR = 2.49, *p* > 0.05) and obese (OR = 4.63, *p* < 0.001). By province, women from KwaZulu-Natal (OR = 0.27, *p* < 0.01) and Gauteng (OR = 0.40, *p* < 0.05) were three and four times less likely to be associated with underweight compared to women in the Western Cape province in urban and rural areas. Furthermore, women from Northern Cape were two times (OR = 1.08, *p* > 0.05) less likely to be underweight compared to those from Western Cape (Table 4).

Regarding weight, women whose weight was above the national average (74.1 kg) were 11,774 times more likely to be obese than those whose weights were below average in urban and rural areas (Table 4). Breastfeeding women were two times more likely to be underweight than non-breastfeeding women in urban and rural areas. Women with long working hours were more likely to be overweight (OR = 1.51, *p* < 0.001) and obese (OR = 1.87, *p* < 0.001) in urban and rural areas, while those currently living with a partner were more likely to be overweight (OR = 1.50, *p* < 0.01) than those who were not, in both urban and rural areas. Women who reported that they smoked cigarettes were found to be more likely to be underweight (OR = 2.65, *p* < 0.01) than those who did not smoke cigarettes, in both urban and rural areas (Table 4).

### 3.5. Multivariate Analysis

To examine the effects of body weights of women and factors by urban–rural variations, we conducted two multivariate analyses using multilevel logistic regression analysis (mixed effect) and ICC. In the first, we estimated the gross effects of the factors influencing body weight among women of childbearing age (Table 5); in the second, we estimated the net effects of the ICC of body weight controlling for other covariates with respect to urban–rural variations in Nigeria and South Africa (Table 6). In Nigeria, the findings of the multilevel regression analysis (models of body weight and factors) indicated that body weight for age significantly decreased the odds of underweight among women of childbearing age 40–44 years (OR = 0.83, *p* < 0.01). Equally, body weight for increasing age significantly increased the odds of being overweight among women aged 45–49 (OR = 7.51, *p* < 0.001) by urban–rural variations (Table 5).

In the full models (overweight), women with primary education (OR = 1.22, *p* < 0.01) had significantly increased odds of being overweight by urban–rural variation (Table 5). On the other hand, women with increased education (secondary (OR = 1.86, *p* < 0.05) and tertiary (OR = 2.45, *p* < 0.001)) were significantly associated with increased odds of obesity by urban–rural variation in Nigeria (Table 5). For employment status, women who were self-employed (OR = 1.37, *p* < 0.001) were significantly associated with increased odds of underweight, while employed women (OR = 13.59, *p* < 0.001) had significantly increased odds of obesity by urban–rural variation (Table 5). For the wealth index, poorer (OR = 1.82, *p* < 0.001), average (OR = 1.99, *p* < 0.01), richer (OR = 2.89, *p* < 0.001), and richest (OR = 4.04, *p* < 0.001) women were significantly associated with increased odds of overweight (Table 5).

By geographical zone, women in the North east (OR = 3.16, *p* < 0.001) had significantly increased odds of being underweight by urban–rural variation compared to women from North central. Women from South east (OR = 1.21, *p* < 0.001) and South south (OR = 1.38, *p* < 0.001) were significantly associated with increased odds of overweight compared to women from North central (Table 5). Conversely, women with height above average (OR = 1.07, *p* > 0.05) had increased odds of being underweight in the urban–rural variation compared to those with height below average, while women with weight above average (OR = 36169.13, *p* < 0.001) were significantly associated with higher odds of obesity by urban–rural variations (Table 5). For contraceptive use, the folkloric method (OR = 1.32, *p* > 0.05) increased the odds of being underweight among women by urban–rural variations compared to those not using contraceptive methods, while traditional (OR = 1.79, *p* < 0.001, OR = 2.23, *p* < 0.001) and modern methods (OR = 1.61, *p* < 0.01, OR = 1.27, *p* < 0.05) were significantly associated with increased odds of overweight and obesity among women of childbearing age by urban–rural variations compared to those women not using any contraceptive method (Table 5).

On the other hand, breastfeeding (OR = 1.27, *p* < 0.001) was significantly associated with underweight compared to those women who were not breastfeeding. Additionally, among women currently living with their partner, those who reported ‘yes’ were significantly associated with higher odds of obesity (OR = 1.05, *p* < 0.001). Women with long working hours (OR = 1.04, *p* > 0.05) had increased odds of being underweight compared to those who did not have long working hours. Women who smoked cigarettes (OR = 2.62, *p* > 0.05) had increased odds of obesity compared to women who did not smoke cigarettes (Table 5). Furthermore, the standard deviation showed that there was high variation by residence type (urban–rural), with over 20% of underweight women, over 30% of overweight women, and 60% of obese women in Nigeria (Table 5). Although decreased odds of underweight were seen among women, higher odds of variations of overweight and obesity were found to be increased among women by residence type (urban–rural) (Table 5).

However, Table 6 shows that there existed significant associations in body weight and factors among women of childbearing age in urban–rural variations in South Africa. In the full models (obesity model), increasing age (20−49 years) was associated with higher odds of obesity. Women with primary (OR = 2.72, *p* < 0.05), secondary (OR = 6.64, *p* < 0.001), and tertiary (OR = 2.92, *p* < 0.05) education were significantly associated with higher odds of overweight by urban–rural variations. For employment status, women who were self-employed (OR = 11.98, *p* < 0.01) and employed (OR = 161820.60, *p* < 0.001) were significantly associated with higher odds of obesity compared to unemployed women by urban–rural variations. For wealth index, having increasing wealth status, being poorer (OR = 4.91, *p* > 0.05), and being average (OR = 8.82, *p* > 0.05) increased the odds of being underweight compared to women in the poorest wealth index by urban–rural variations. For provinces, women from Eastern Cape (OR = 5.25, *p* < 0.001), KwaZulu-Natal (OR = 4.87, *p* < 0.001), North west (OR = 3.05, *p* < 0.01), Mpumalanga (OR = 2.63, *p* < 0.05), and Limpopo (OR = 4.56, *p* < 0.001) were significantly associated with higher odds of being overweight by urban–rural variations (Table 6).

Similarly, for the urban–rural variation, being resident in Eastern Cape (OR = 144.32, *p* < 0.001), Free State (OR = 1477.11, *p* < 0.001), KwaZulu-Natal (OR = 1717.30, *p* < 0.001), Northwest (OR = 1626.01, *p* < 0.001), Gauteng (OR = 14.51, *p* < 0.05), and Mpumalanga (OR = 58.84, *p* < 0.01) significantly increased the odds of obesity among women of childbearing age. For weight, being above average was significantly associated with increased odds of overweight (OR = 569.35, *p* < 0.001) and obesity (OR = 2.9 × 10^19^, *p* < 0.001) by urban–rural variations. Having seven or more children (OR = 273.08, *p* < 0.01) was significantly associated with higher odds of underweight by urban–rural variations. For the contraceptive method, the modern method (OR = 1.43, *p* < 0.05) was significantly associated with higher odds of being overweight among women of childbearing age by urban–rural variations. Similarly, breastfeeding women were significantly associated with higher odds of overweight (OR = 1.69, *p* < 0.05) and obesity (OR = 22.68, *p* < 0.01) by urban–rural variations (Table 6).

Furthermore, women currently living with their partner were found to be significantly associated with higher odds of being overweight (OR = 1.81, *p* < 0.01), while women with long working hours (OR = 5.44, *p* > 0.05) had higher odds of being underweight compared to those not having long working hours by urban–rural variations. For cigarette smoking, women who smoked had increased odds of being either underweight (OR = 2.14, *p* < 0.01) or obese (OR = 145.99, *p* < 0.001) by urban–rural variations. Furthermore, the standard deviation in Table 6 indicates high variation by residence type (urban–rural), with over 16% of underweight women, 29% of overweight women, and 60% of obese women in South Africa (Table 6). Although decreased odds of underweight were seen among women, higher odds of variations of overweight (29%) and obesity (48%) were found to be increased among women by residence type (urban–rural).

Table 7 reports the findings of the multilevel model of the intracluster correlation coefficient and the respective 95% CI for Nigeria and South Africa. The median ICC (0.0102) for underweight body weight categories in South Africa was lower compared to Nigeria (0.0127). For other body weight categories, the ICC medians (overweight—0.0289 and obese—0.1040) for Nigeria were moderately higher compared with the ICC medians for South Africa (overweight—0.0271 and obese—0.0819). The study findings revealed that Nigeria had the highest ICC for BMI, revealing an increase in body weights among women of childbearing age. However, in South Africa, with the existing prevalence of body weight categories, some remarkably significant changes were observed as the preventive approaches and interventions put in place are having an impact on BMI among South African women of childbearing age. However, in Nigeria, women are still plagued by ignorance, poor knowledge, and perceptions of health implications of malnutrition.

## 4. Discussion

In this paper, we used data from two demographic health surveys (DHS), i.e., the 2018 NDHS and 2016 SADHS, to identify the urban–rural variations between women’s body weight and individual-level factors in Nigeria and South Africa. According to this study, the overall prevalence rates for overweight and obesity among women of childbearing age were 28.2% and 44.9% in South Africa. Similarly, the overall prevalence of overweight and obesity among women in South Africa in our study is comparable to evidence from the SADHS [12] and existing studies conducted in South Africa among black men and women [36,37]. The findings showed that there is a higher prevalence of overweight and obesity among women aged between 30 and 59 years in South Africa, while the prevalence of overweight and obesity from other piloted studies conducted outside South Africa was lower [38,39,40,41,42,43]. The high prevalence of overweight and obesity obtained from previous South African studies piloted may in part be due to widespread sedentary lifestyles and a surge in processed food outlets, largely reflective of a trend across many African settings [36,37,44].

On the other hand, a prevalence of overweight (20.2%) and obesity (11.4%) was obtained among women of childbearing age in Nigeria. This finding is not inconsistent with previous community surveys conducted in Nigeria, as well as a systematic review of Nigerian studies on overweight and obesity, where an increasing prevalence trend of overweight (25.5–45.3%) and obesity (19.8–40.2%) was obtained [45,46,47]. A higher prevalence of overweight and obesity was marked among women of reproductive age obtained from previous Nigerian studies piloted [9,10,11], which may be linked to cultural lifestyles, dietary choices, and sedentary lifestyles. However, a higher prevalence of underweight (9.0%) and normal weight (59.4%) was found among women in Nigeria compared to women who are underweight (1.8%) or have a normal weight (25.1%). This finding supports earlier research which showed the coexistence of both undernutrition and overnutrition in the same population, which over time has resulted in significant morbidity and mortality [46,47].

Research indicates that women from Nigeria residing in urban areas have a higher possibility of being overweight/obese, while rural women were found to be more likely to be either underweight or overweight. The findings of this study are consistent with previous studies from South Asian [48,49,50,51,52,53] and sub-Saharan Africa countries [1,2,45,54], which showed that women from rural households were at a higher risk of being underweight or overweight compared with their urban counterparts. The likely reason for the urban–rural differences in body weights of Nigerian women is due to a high level of financial constraints, shortage of job prospects, poor availability of healthcare services, and a sedentary and unhealthy dietary lifestyle (such as high intake of extremely caloric foods and poor consumption of fruits and vegetables) [55,56,57]. Similarly, this finding is in line with studies in India and Nepal, which found that women in rural households were either underweight or overweight, as rural women are vulnerable to malnutrition [48,49]. This is also consistent with other study reports in low- and middle-income countries [54]. This might be due to the fact that residing in rural areas is one of the determinant factors significantly associated with a high prevalence of underweight in this study and in other studies [58,59,60,61].

Furthermore, the findings of this study propose that long-term intervention to reduce the burden and health implications of being underweight/overweight among rural households should be aimed at women of low socio-economic class and who are from the poorest households. Moreover, in our study, which is consistent with other studies [62,63], there was a high prevalence of overweight and obesity among urban South African women. The prevalence of overweight/obesity was comparatively greater in urban than in rural settings [12]. Thus, multiple factors are likely to contribute to the increased prevalence of overweight/obesity in urban populations, including the presence of modern communication facilities, increased availability of technology, easy access to energy-rich food, reduced levels of physical activity, and adoption of a sedentary lifestyle [57,58]. Studies revealed a higher prevalence of overweight and obesity among South African women of childbearing age compared with women from other African countries [43,59,60]; this may be as a result of nutritional and epidemiological shifts, which are determined by factors such as demographic variations, rising earnings, suburbanization, unhealthy lifestyles, and consumption of highly processed diets, which appear to be strong drivers of a prevailing overweight and obesity epidemic among women of childbearing age in South Africa.

As expected, and consistent with other studies, our descriptive bivariate findings showed that there were significant differences in terms of body weight and socio-demographic factors in urban and rural populations in this study. The odds of being underweight among Nigerian women of childbearing age were increased by being self-employed, as well as being from the North east and North west geopolitical zones; breastfeeding was also significantly associated. This is consistent with other studies [27,28,29,59] and could be due to differences in educational status, food security, and access to information regarding nutrition education to address the prevalent underweight burden, especially among Nigerian women in rural grassroots communities. Moreover, in South Africa, our study findings showed that women of childbearing age who were underweight were more likely to belong to poorer wealth households, reside in the Northern Cape and Limpopo provinces, have height above average, be currently breastfeeding, and smoke cigarettes. These study findings are in line with other studies conducted in South Asian [49,51,52] and other African countries [39,40,59]. Studies have documented that the upturn in women’s employment is an important contributing factor to the direct causes of undernutrition such as feeding practices and ill health, and more distinct bases of undernutrition (such as income, food security, and education) have a greater prospect to improve women’s nutritional status [64,65,66].

However, there were significant differences among overweight women and sociodemographic factors, such as increasing age (30–34 years; 35–39 years), secondary education, middle wealth index, divorced/separated, regional differences (South and South west zones), weight above average, contraceptive use (traditional and modern), and having more than four children. Similarly, significant differences were found among women who were obese, with socio-demographic factors, such as increasing age (40–44 years; 45–49 years), higher education, employed, higher wealth index, married/widowed, regional differences (South east, South south, and South west zones), height above average, weight above average, having more than four children, contraceptive use (traditional/modern), and cigarette smoking. Several studies (cross-sectional and longitudinal) have demonstrated the influence of socio-demographic factors with studies from sub-Saharan Africa [1,2,67], and this is consistent with findings from studies conducted in Nigeria [9,13,22].

In South Africa, the risk of being overweight and/or obese was also higher among women with increasing age (20–24; 25–29; 30–34; 35–39; 40–44; 45–49 years), higher education, employed, residing in higher wealth households, and presently married. For South African women, weight above average, having more than four children, and contraceptive use (modern method) were also significantly associated with obesity. Residing in urban settings (Eastern Cape, Free State, and Mpumalanga provinces) was associated with a higher risk of being overweight. Research indicates that women who are currently breastfeeding, are living with a partner, and have long working hours showed significant differences. Thus, women who are employed with long working hours were found to be obese. The results of this study are consistent with previous studies from Ghana [58,60], South Asian [51,52,55], and sub-Saharan Africa countries [2,45].

In Nigeria and South Africa, overweight and obesity are often associated with affluence, and educational attainment is often used as a substitute indicator for socio-economic status. Therefore, it is not unexpected that the bivariate findings of this study showed that odds of overweight and obesity greatly increased as educational attainment increased among this study population. Women with tertiary education were two times more likely to be overweight or obese compared to those with no education. Similarly, women with higher education had progressively higher odds of being overweight/obese compared to those with no education. This finding is in agreement with the results from other studies conducted in Ethiopia [6,34,40,67] and in Asian countries [48,50,51,52,53]. A likely explanation for this could be that, with higher education, women are more likely to earn a better income, and this makes it easier to adopt a more westernized way of life, which has been reported by several studies to be increasing the obesity epidemic in Nigeria [21,22] and South Africa [15,19].

From the multi-level findings of this study, older women were more associated with increased likelihood of overweight/obesity in Nigeria, and this supports previous studies showing that, as women get older, they face greater risks of being overweight/obese [10,16]. Women with secondary or higher education had higher odds of being overweight, while being employed was identified as a predictor of overweight/obesity in Nigeria and South Africa. Our findings also agree with these earlier studies, as socio-economic status is one of the major predictors of overweight/obesity in both countries [51,64,68,69]. In addition, research indications have revealed that higher education attainment is associated with the better health status of the community, owing to an improvement in socio-economic status [68,69], health literacy and health behaviors [65,66], and self-control and empowerment [51,64,69]. This is not always the case in less developed and developing countries such as Nigeria and South Africa, where those with higher education are more likely to be overweight or obese [37,38,41,44].

Consistent with other studies conducted in other African countries, we found that South African women of childbearing age who attained secondary or higher education were more likely to be overweight/obese, similar to studies from Ghana [58,60], Bangladesh [52,53], and Ethiopia [6,40,67]. The likely reasons for this study finding in Nigeria and South Africa are that women with higher education are more likely to have a higher socio-economic status and material resources and have ready access to energy-dense nutrition and sugary beverages, as well as a more sedentary paying job. Women from wealthier households, having weight above average, and presently living with partner had increased odds of being overweight or obese, among women from Nigeria [9,22] and South Africa [14,15]. Moreover, as the number of children increases, the odds of being overweight or obese were decreased, and smoking of cigarettes also increased the likelihood of being underweight among South African women. The results of this study are inconsistent with previous studies from Ghana [58,60], South Asian [51,61], and sub-Saharan Africa countries [2,45].

In Nigeria, women who are currently breastfeeding were found to have higher odds of being underweight compared to women in South Africa, who had lower odds of being overweight or obese when breastfeeding. Thus, weight loss or gain can be achieved during breastfeeding. Studies have shown that calories from fat cells stored in the body to produce milk are burned during breastfeeding [27,59,70]. Furthermore, weight can be gained during breastfeeding if nursing mothers do not adopt healthy choices of dietary intake with increasing calories, rather than the necessary macro- and micronutrients [63,70]. This specific biochemical process depends on genetic factors, metabolism, and the hormone prolactin, which stimulates appetite, and breastfeeding predisposes nursing mothers to gain rapid weight during breastfeeding [28,29]. Furthermore, lack of sleep during breastfeeding may cause increased appetite and cravings for high-fat and high-calorie foods, stimulating weight gain among nursing mothers.

Our results showed variations in the way that regions of residence predict underweight and overweight in Nigeria. For instance, the North east region was consistently associated with increased odds of being underweight, whereas the South east and South south regions were consistently associated with increased odds of being overweight. These results are also in agreement with earlier studies which highlighted the influence of locational factors on body weight categories [11,16,22]. In South Africa, women residing in Eastern Cape, Free State, KwaZulu-Natal, Northwest, Gauteng, and Mpumalanga were more associated with increased likelihood of obesity. This finding supports the higher odds of overweight and obesity by province reported by the 2016 South African Demographic Health Survey [12,15]. These results are also in agreement with earlier studies which highlighted the influence of locational factors on body weight categories [58,62]. The reason for this could also be due to dissimilarities in the socio-cultural and environmental practices and the dynamics prevailing in different provinces/geopolitical zones. These varying factors could lead to changes in the dimensions of culture and traditional beliefs, as well as the socio-economic status of the population, all of which usually have an intense influence on cultural interpretations and body weight [36,62,71,72].

However, the ICC for body weight and its associated factors among Nigerian and South African women of childbearing age varied substantially, from a minimum of underweight of 0.0102 in South Africa to a maximum of 0.0127 for Nigeria. Additionally, variances can be found in the overweight and obesity categories in Models 2 and 3 with respect to Nigeria and South Africa (Table 4). These findings indicated that universal strategies to control overweight and obese body categories may not consistently show effective outcomes in both countries [17,34,62]. For instance, some strategic interventions or preventive approaches regarding the problem of body weight in Nigeria may not be equally effective in South Africa. Therefore, each country should modify the World Health Organization’s (WHO) or other international strategies according to the country’s needs in terms of clustering in geographical areas (ICC). Countries with a low ICC, such as South Africa as found in this study, should consider giving more emphasis to an entire population approach [34,62]. Countries such as Nigeria with a high ICC should consider adding targeted population approaches to the aggregate population approach [34,62]. These approaches should be directed to identify those households with low socio-economic status that are eligible, and strict targeting should be a priority for any countries with variations in ICC.

Furthermore, these approaches to address high ICC variations for different countries have the potential for implementing public health interventions that are aimed at increasing healthy behavioral factors by targeting those geographic clusters with poor or unhealthy lifestyles. Similarly, to improve health behavioral factors, a targeted population approach should be implemented to curb unhealthy lifestyles such as low physical activity, unhealthy dietary intake and enactment of government policies to control advertisement and marketing of junk or fast food among populations in high-risk areas [73,74,75]. Importantly, most developed countries across Europe, China and the United States of America have made efforts to place stringent advertisement controls, most especially directed at high-risk persons with overweight and obesity health concerns and the provision of supplemental nutrition assistance programs, to encourage adopting healthy lifestyle behaviors. In addition, determining the predictors of body weight and its associated factors using the intracluster correlation coefficient (ICC) may help to modify the public health interventions for body weight, socio-demographic, and behavioral factors according to geographic variations, targeting interventions in Nigeria and South Africa.

### Strengths and Limitations

This study had several major strengths and limitations. Firstly, to the best of the authors’ knowledge, this is the first cross-sectional survey and nationally representative health demography and medical sociology study researching body weight and its individual-level factors among women of childbearing age by urban–rural variations in Nigeria and South Africa. Secondly, the data analysis was basically conducted to determine the association based on predictor likelihood and not a measure of causality; however, insight can be gained from comparing the 2018 DHS dataset from Nigeria and 2016 DHS dataset from South Africa to improve the study’s generalizability to other settings or populations. Thirdly, to the best of the authors’ knowledge, this is the first time that multi-level regression models were aimed at elucidating the predicting factors of the possibility of being underweight, overweight, or obese compared to normal weight in Nigeria and South Africa. A final strength is the practical application of employing the ICC of multilevel analytical methods in determining the magnitude of exposure of women of childbearing age in different countries to the characteristics associated with body weight.

There were some limitations, however, that need to be highlighted. Firstly, owing to the cross-sectional nature of the studies (2018 NDHS and 2016 SADHS), we cannot draw causal inferences from the findings. Secondly, possible bias in reporting body weights (underweight, normal, overweight, and obese) could have occurred related to individuals’ recall of behavioral factors, especially lifestyle factors. Thirdly, self-reported data on height and weight of the individuals were used in generating individual BMI, and possible errors might have occurred when recording the weight and height of the respondents.

## 5. Conclusions

Our findings are in support of the literature that overweight/obesity is more prevalent among urban women in South Africa than in Nigeria, and rural women were more underweight in Nigeria than in South Africa, suggesting prevailing poor nutritional transition and epidemiological shifts. Other significant drivers of underweight in Nigeria were being unemployed, regional differences, and breastfeeding, whereas, in South Africa, factors such as poorer households, provincial differences, and cigarette smoking were associated with being underweight. Meanwhile, factors such as increasing age, higher education, richer/rich wealth status, weight above average, and traditional/modern contraceptive use were associated with increased risk of overweight and obesity in Nigeria and South Africa. This study revealed the influence of regional/provincial differences on unhealthy body weights; most importantly, overweight/obesity has the ability to fluctuate according to women’s different socio-economic and socio-cultural backgrounds, as well as genetic factors. An indication of probable locational factors (urban or rural) among women with increasing age, breastfeeding, or having poor livelihoods was perceived in Nigeria and South Africa. In Nigeria, the progressive tendency of overweight/obesity burden is becoming alarming with higher incidence, and concerted efforts are required to enlighten women to adopt healthy lifestyles in fighting overweight/obesity health concerns and its concomitant health risks.

Similarly, in Nigeria and South Africa, adolescent and young adult women deserve the response of immediate interventions to increase their knowledge and awareness of health-promoting behaviors, since being underweight in this age group is associated with increased morbidity and maternal deaths in the near future. Thus, individual-level factors are key concerns for interactions among researchers, policymakers, and other stakeholders to integrate effective and identified preventive approaches aimed at each country in promoting healthy body weights. Our findings will go a long way in helping decision-makers to improve health and prevent diseases associated with body weight, by simultaneously targeting those clustered areas at risk and women at risk to improve healthy body weight in Nigeria and South Africa. Further studies are needed to determine the underlying behavioral predictors for underweight and overweight/obesity such as low dietary intake, alcohol consumption, tobacco use, and sedentary lifestyles using a qualitative methodology and longitudinal surveys in Nigeria and South Africa for comparison.

## 6. Contribution to the Field

Body weight (being underweight, overweight, or obese) is a significant health and social problem, reaching its peak in South Africa and Nigeria. This is as a result of persistent obesogenic lifestyle changes with environmental impact on Nigerian and South African populations. The contemporary facts identify the prevalence and predictors of body weight, with the ultimate objective of providing health strategic programmes to increase health-promoting behaviors of women of childbearing age in Nigeria and South Africa. These findings have public health implications for Nigeria and South Africa, as actions are needed to combat unhealthy body weights among women in Nigeria and South Africa. In general, a high prevalence of underweight and overweight/obesity among women and the significant differences in individual-level factors such as education, employment, and socio-economic status call for more investment in health literacy and behavioral change. The key predictors of body weight and its associations with various individual factors in clustering areas of Nigeria and South Africa highlight the need for more tailored cultural and community interventions, specific to each country, to slow down the forms of a malnutrition epidemic. In addition, a high prevalence of overweight/obesity among South African and Nigerian women and the positive associations with increasing education, employment, and highest wealth index, embedded in negative health implications call for sensitization of how to change people’s attitudes, beliefs, and perceptions of social norms of lifestyles. Furthermore, the existing study contributed by distinguishing body weight and individual-level factors in multi-level analysis and identifying the implications of these differences for health demographers and medical sociologists, as well as public health researchers.

## Figures and Tables

**Figure 1 ijerph-19-00125-f001:**
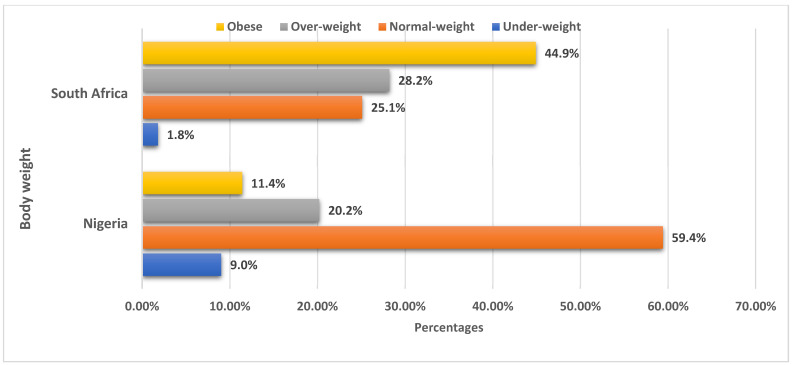
Overall prevalence of body weight among women of childbearing age (15–49 years) by country. Source: NDHS, 2018 and SADHS, 2016.

**Figure 2 ijerph-19-00125-f002:**
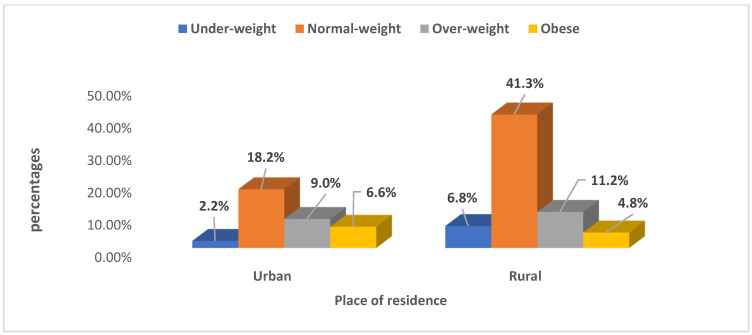
Body weight among women of childbearing age (15–49 years) by urban-rural variations in Nigeria. Source: NDHS, 2018.

**Figure 3 ijerph-19-00125-f003:**
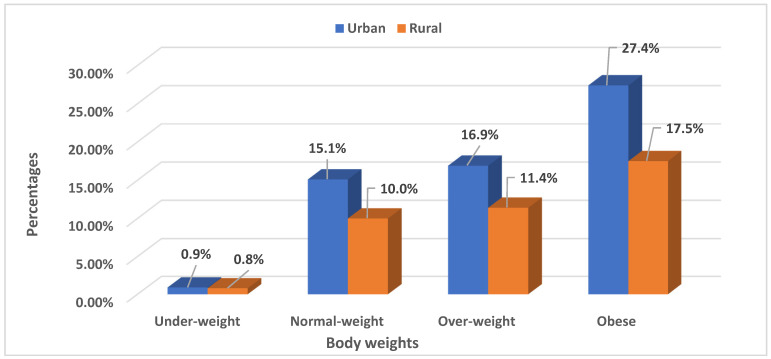
Body weight among women of childbearing age (15–49 years) by urban–rural variations in South Africa. Source: SADHS, 2016.

**Table 1 ijerph-19-00125-t001:** Summary of independent variables measured in this study.

S/No	Variable Name	Categorization
1	Age	15–19 = 1; 20–24 = 2; 25–29 = 3; 30–34 = 4; 35–39 = 5; 40–44 = 6; 45–49 = 7
2	Place of residence	urban = 1; rural = 2
3	Woman education	no education = 1; primary = 2; secondary = 3; higher education = 4
4	Employment status	unemployed = 1; self-employed = 2; employed = 3
5	Wealth index	poorest = 1; poorer = 2; average = 3; richer = 4; richest = 5
6	Marital status	single = 1; married = 2; cohabiting = 3; widowed = 4; divorced/separated = 5
7	Geopolitical zone	North central = 1, North east = 2, North west = 3, South east = 4, South west = 5, South south = 6
8	Province	Western Cape = 1; Eastern Cape = 2; Northern Cape = 3; Free State = 4; KwaZulu-Natal = 5; North West = 6; Gauteng = 7; Mpumalanga = 8; Limpopo = 9.
9	Height	below average = 1; above average = 2
10	Weight	below average = 1 and above average = 2
11	Children ever born	1–3 = 1; 4–6 = 2; 7+ = 3
12	Contraceptive method	none = 1; folkloric = 2; traditional = 3; modern = 4
13	Breastfeeding	no = 1; yes = 2
14	Living with partners	no = 1; yes = 2
15	Long working hours	no = 1; yes = 2
16	Cigarette smoking	no = 1; yes = 2

Source: Authors’ compilation.

**Table 2 ijerph-19-00125-t002:** Socio-demographic, geographical, and behavioral characteristics of women in Nigeria and South Africa.

Characteristics	Nigeria		South Africa	
	Frequency (*n*)	Percentage (%)	Frequency (*n*)	Percentage (%)
**Age group**				
15–19	1489	1.2	194	1.4
20–24	8568	6.8	1123	7.9
25–29	19,202	15.2	2188	15.5
30–34	24,026	19.0	2687	19.0
35–39	26,641	21.1	2646	18.7
40–44	23,080	18.2	2628	18.6
45–49	23,532	18.6	2675	18.9
**Residence**				
Urban	48,493	38.3	9010	63.7
Rural	78,045	61.7	5134	36.3
**Education**				
No education	65,030	51.4	512	3.6
Primary	23,683	18.7	1999	14.1
Secondary	29,876	23.6	10,219	72.3
Tertiary	7948	6.3	1413	10.0
**Employment status**				
Unemployed	228	0.2	10,656	75.3
Self-employed	91,157	92.7	1596	11.3
Employed	7000	7.1	1892	13.4
**Wealth index**				
Poorest	19,586	23.4	3109	22.0
Poorer	29,135	23.0	3107	22.0
Average	25,754	20.4	3155	22.3
Richer	23,326	18.4	2643	18.7
Richest	18,736	14.8	2129	15.1
**Marital status**				
Single	1437	1.14	5587	39.5
Married	113,851	90.0	4971	35.1
Cohabiting	2861	2.26	2311	16.3
Widowed	5215	4.12	556	3.9
Divorced/separated	3173	2.51	718	5.1
**Geopolitical zone**				
North central	16,460	13.0	-	-
Northeast	22,686	17.9	-	-
Northwest	47,751	37.7	-	-
Southeast	12275	9.7	-	-
Southwest	11,228	8.9	-	-
South south	16,138	12.8	-	-
**Provinces**				
Western Cape	-	-	1547	10.9
Eastern Cape	-	-	1648	11.7
Northern Cape	-	-	289	2.0
Free State	-	-	681	4.8
KwaZulu–Natal	-	-	2553	18.1
Northwest	-	-	1057	7.5
Gauteng	-	-	3673	26.0
Mpumalanga	-	-	1235	8.7
Limpopo	-	-	1459	10.3
**Height**				
Below average	19,664	48.3	2334	45.1
Above average	21,016	51.7	2843	54.9
**Weight**				
Below average	27,592	59.3	3019	52.2
Above average	18,979	40.8	2763	47.8
**Children borne**				
1–3	27,958	22.1	9746	68.9
4–6	48,973	38.7	3832	27.1
7^+^	49,606	39.2	566	4.0
**Contraceptive method**				
None	105,955	83.7	6186	43.7
Folkloric	900	0.7	0	0.0
Traditional	4430	3.5	35	0.3
Modern	15,253	12.1	7923	56.0
**Breastfeeding**				
No	85,963	67.9	12,516	88.5
Yes	40,575	32.1	1628	11.5
**Living with partner**				
No	8083	6.9	1229	17.7
Yes	108,630	93.1	5735	82.4
**Long working hours**				
No	32,692	25.8	8376	59.2
Yes	93,846	74.2	5770	40.8
**Cigarette smoking**				
No	126,272	99.8	6702	94.8
Yes	266	0.2	371	5.3
**Total**	126,538	100.0	14,144	100.0

Source: Nigeria Demographic Health Survey, 2018 and South Africa Demographic Health Survey, 2016.

**Table 3 ijerph-19-00125-t003:** Bivariate analysis showing adjusted odds ratio of body weights and associated factors among women of childbearing age by urban–rural variations in Nigeria.

Factors	Underweight		Overweight		Obese	
	AOR	95% CI	AOR	95% CI	AOR	95% CI
**Age group**						
15–19	1.00		1.00		1.00	
20–24	0.92	0.75–1.13	1.80	0.72–4.46	13.56 ***	8.91–20.63
25–29	0.75 ***	0.70–0.79	3.13 *	1.21–8.11	31.66 ***	14.57–68.78
30–34	0.86 *	0.76–0.96	3.92 **	1.81–8.48	48.99 ***	36.30–66.10
35–39	0.74 ***	0.64–0.83	5.02 ***	2.06–12.19	67.22 ***	33.65–134.24
40–44	0.61 ***	0.55–0.68	4.20 **	1.70–10.32	71.98 ***	35.75–144.91
45–49	0.69 ***	0.62–0.77	4.99 **	1.74–14.25	98.01 ***	41.65–230.62
**Education**						
No education	1.00		1.00		1.00	
Primary	0.49 ***	0.36–0.66	1.98 ***	1.79–2.16	2.76 ***	1.96–3.91
Secondary	0.45 ***	0.35–0.58	2.36 ***	1.87–2.96	3.91 ***	2.10–7.27
Tertiary	0.34 **	0.17–0.65	3.43 ***	2.69–4.35	7.30 ***	4.99–10.66
**Employment status**						
Unemployed	1.00		1.00		1.00	
Self-employed	1.49 ***	1.20–1.84	1.46	0.25–8.35	4.70 **	1.76–12.56
Employed	0.97	0.48–1.99	2.19	0.32–14.88	9.02 ***	3.09–26.33
**Wealth index**						
Poorest	1.00		1.00		1.00	
Poorer	0.71 *	0.51–0.97	1.78 ***	1.75–1.82	3.39 ***	2.09–5.48
Average	0.49 ***	0.41–0.57	2.51 ***	2.18–2.88	5.32 ***	4.12–6.89
Richer	0.43 ***	0.38–0.47	4.24 ***	4.14–4.34	12.49 ***	9.92–15.73
Richest	0.34 ***	0.27–0.42	6.49 ***	6.18–6.82	29.12 ***	23.24–36.48
**Marital status**						
Single	1.00		1.00		1.00	
Married	1.01	0.40–2.52	1.02	0.60–1.73	1.18 ***	1.08–1.28
Cohabiting	0.58 ***	0.52–0.65	0.94	0.45–1.98	1.23	0.73–2.07
Widowed	0.61 *	0.38–0.96	0.89 *	0.81–0.98	1.58 ***	1.23–2.02
Divorced/separated	1.16	0.57–2.33	1.40 **	1.14–1.71	1.53	0.65–3.59
**Geopolitical zone**						
North central	1.00		1.00		1.00	
Northeast	2.91 ***	2.34–3.61	0.64 **	0.47–0.86	0.63 *	0.42–0.92
Northwest	1.48 ***	1.22–1.77	0.62 ***	0.52–0.74	0.40	0.16–1.00
Southeast	0.75	0.34–1.59	1.36	0.94–1.98	1.55 *	1.03–2.32
South south	0.69	0.41–1.15	1.99 ***	1.52–2.61	2.45 ***	2.13–2.80
Southwest	0.99	0.45–2.17	1.37 ***	1.15–1.61	1.40 ***	1.32–1.48
**Height**						
Below average	1.00		1.00		1.00	
Above average	1.11	0.92–1.33	1.07	0.98–1.16	1.32 **	1.12–1.54
**Weight**						
Below average			1.00		1.00	
Above average			50.53 ***	33.87–75.35	1783.32 ***	530.8–5991.1
**Children borne**						
1–3	1.00		1.00		1.00	
4–6	0.89 ***	0.86–0.92	1.22 ***	1.14–1.29	1.50 ***	1.39–1.61
7^+^	1.04	0.89–1.23	0.98	0.76–1.26	0.98	0.91–1.08
**Contraceptive method**						
None	1.00		1.00		1.00	
Folkloric	1.05	0.65–1.66	1.33	0.91–1.93	0.63	0.14–2.79
Traditional	0.64 **	0.49–0.84	2.62 ***	1.92–3.58	3.36 ***	2.81–4.01
Modern	0.66 ***	0.58–0.74	2.03 ***	1.89–2.17	2.04 ***	1.86–2.23
**Breastfeeding**						
No	1.00		1.00		1.00	
Yes	1.26 ***	1.16–1.37	0.71 ***	0.710–0.714	0.45 ***	0.37–0.55
**Living with partner**						
No	1.00		1.00		1.00	
Yes	1.55	0.93–2.56	0.77	0.57–1.05	0.80	0.54–1.17
**Long working hours**						
No	1.00		1.00		1.00	
Yes	0.87 ***	0.81–0.93	1.50	1.07–2.08	1.87	0.90–3.88
**Cigarette smoking**						
No	-		1.00		1.00	
Yes	-	-	0.76	0.09–6.59	4.52 ***	3.43–5.93

Normal weight was the reference group, adjusted for all variables in the column (reference category = 1.00). * *p* < 0.05, ** *p* < 0.01, *** *p* < 0.001. OR (odds ratio); AOR (adjusted OR).

**Table 4 ijerph-19-00125-t004:** Bivariate analysis showing adjusted odds ratio of body weight and its associated factors among women of childbearing age by urban–rural variations in South Africa.

Factors	Underweight		Overweight		Obese	
	AOR	95% CI	AOR	95% CI	AOR	95% CI
**Age group**						
15–19	1.00		1.00		1.00	
20–24	3.73	0.58–23.85	3.44 ***	1.89–6.24	2.74 **	1.44–5.21
25–29	3.95	0.63–24.41	4.01 ***	2.24–7.15	4.60 ***	2.48–8.53
30–34	3.27	0.52–20.51	4.77 ***	2.67–8.52	7.89 ***	4.26–14.59
35–39	7.22 *	1.18–44.01	4.34 ***	2.41–7.79	12.31 ***	6.65–22.76
40–44	1.55	0.22–10.72	5.68 ***	3.17–10.17	11.51 ***	6.21–21.30
45–49	1.59	0.22–11.28	6.30 ***	3.50–11.33	16.04 ***	8.65–29.77
**Education**						
No education	1.00		1.00		1.00	
Primary	1.67	0.07–36.65	1.57 *	1.05–2.34	0.99	0.71–1.37
Secondary	1.41	0.25–7.65	1.81 **	1.25–2.59	1.21	0.90–1.63
Tertiary	0.11 ***	0.04–0.29	2.14 ***	1.37–3.35	1.56 *	1.07–2.28
**Employment status**						
Unemployed	1.00		1.00		1.00	
Self-employed	0.83	0.28–2.43	1.47 **	1.16–1.85	1.06	0.84–1.33
Employed	0.35	0.02–5.62	1.95 ***	1.47–2.56	2.59 ***	2.02–3.34
**Wealth index**						
Poorest	1.00		1.00		1.00	
Poorer	2.36 ***	1.56–3.56	0.98	0.80–1.20	1.21 *	1.00–1.46
Average	2.07	0.44–9.66	1.13	0.92–1.39	1.64 ***	1.34–1.99
Richer	1.35	0.53–3.43	1.14	0.92–1.43	1.68 ***	1.34–2.10
Richest			1.33	1.02–1.74	2.48 ***	1.91–3.21
**Marital status**						
Single	1.00		1.00		1.00	
Married	0.47 *	0.25–0.87	1.78	1.48–2.12	2.52 ***	2.14–2.96
Cohabiting	0.60	0.33–1.09	1.20	0.98–1.46	1.04	0.86–1.26
Widowed	2.49	0.93–6.60	2.33	1.42–3.82	4.63 ***	2.98–7.18
Divorced/separated	0.36	0.06–1.95	2.58	1.79–3.71	2.34 ***	1.64–3.31
**Provinces**						
Western Cape	1.00		1.00		1.00	
Eastern Cape	0.61	0.26–1.43	1.08	0.77–1.51	0.78	0.58–1.06
Northern Cape	1.08	0.34–3.38	0.78	0.46–1.3	0.46 **	0.28–0.77
Free State	0.72	0.24–2.18	1.17	0.77–1.79	0.94	0.64–1.38
KwaZulu-Natal	0.27 **	0.10–0.69	0.87	0.63–1.20	0.83	0.62–1.10
Northwest	0.72	0.29–1.78	0.87	0.60–1.25	0.76	0.55–1.06
Gauteng	0.40 *	0.18–0.90	0.95	0.69–1.28	0.73 *	0.55–0.95
Mpumalanga	0.51	0.19–1.34	1.06	0.74–1.51	0.74	0.53–1.02
Limpopo	1.33	0.62–2.85	0.92	0.65–1.29	0.77	0.56–1.05
**Height**						
Below average	1.00		1.00		1.00	
Above average	1.63 *	1.05–2.52	1.15	0.99–1.35	1.05	0.91–1.20
**Weight**						
Below average	1.00		1.00		1.00	
Above average	-	-	329.67	60.95–1783.1	11,773.93 ***	2168.7–63,919.9
**Children borne**						
1–3	1.00		1.00		1.00	
4–6	0.96	0.59–1.57	1.13	0.95–1.33	1.47 ***	1.27–1.71
7 and above	0.88	0.31–2.51	1.09	0.77–1.54	1.00	0.72–1.38
**Contraceptive method**						
None	1.00		1.00		1.00	
Traditional	-	-	0.049	0.01–1.29	0.04 *	0.01–0.72
Modern	1.43	0.93–2.17	1.14	0.98–1.32	1.21 **	1.06–1.38
**Breastfeeding**						
No	1.00		1.00		1.00	
Yes	1.55	0.92–2.60	1.07	0.86–1.31	0.48 ***	0.38–0.59
**Living with partner**						
No	1.00		1.00		1.00	
Yes	2.74	0.77–9.72	1.50 **	1.11–2.01	1.07	0.83–1.37
**Long working hours**						
No	1.00		1.00		1.00	
Yes	1.06	0.68–1.67	1.51 ***	1.29–1.76	1.87 ***	1.62–2.15
**Cigarette smoking**						
No	1.00		1.00		1.00	
Yes	2.65 **	1.40–5.00	0.58	0.41–0.82	0.60 **	0.44–0.81

Normal weight was the reference group, adjusted for all variables in the column (reference category = 1.00). * *p* < 0.05, ** *p* < 0.01, *** *p* < 0.001. OR (odd ratio); AOR (adjusted OR).

**Table 5 ijerph-19-00125-t005:** Multi-level analysis showing the predictors of body weight among women of childbearing age and factors by urban–rural variation in Nigeria.

Factors	Underweight		Overweight		Obese	
	UOR	95% CI	UOR	95% CI	UOR	95% CI
**Age group**						
15–19	1.00		1.00		1.00	
20–24	1.35 **	1.08−1.66	2.61 *	1.05−6.44	1	-
25–29	1.11 **	1.03−1.18	3.94 **	1.75−8.86	0.34	0.07−1.64
30–34	1.39 ***	1.20−1.58	5.18 ***	2.39−11.19	0.37 ***	0.34−0.39
35–39	1.01	0.69−1.45	6.67 ***	3.00−14.82	0.72 ***	0.69−0.76
40–44	0.83 **	0.82−0.83	6.34 **	2.12−18.91	0.72	0.50−1.05
45–49	1.05	0.82−1.35	7.51 ***	2.84−19.48	0.93	0.79−1.08
**Education**						
No education	1.00		1.00		1.00	
Primary	0.48 ***	0.41–0.56	1.22 **	1.07–1.39	1.35 ***	1.33–1.36
Secondary	0.56 ***	0.50–0.63	1.18	0.71–1.94	1.86 *	1.11–3.12
Tertiary	0.400 ***	0.25–0.66	1.40	0.94–2.07	2.45 ***	1.79–3.35
**Employment status**						
Unemployed	1.00		1.00		1.00	
Self–employed	1.37 ***	1.26–1.48	0.82	0.22–3.01	14.49 ***	9.18–22.85
Employed	1.26	0.80–1.98	0.77	0.18–3.20	13.59 ***	4.61–40.06
**Wealth index**						
Poorest	1.00		1.00		1.00	
Poorer	0.95	0.60–1.49	1.82 ***	1.57–2.12	1.31	0.71–2.42
Average	0.71 ***	0.66–0.77	1.99 **	1.40–2.81	1.27	0.73–2.21
Richer	0.90	0.57–1.43	2.89 ***	1.62–5.15	2.08	0.81–5.28
Richest	0.92	0.51–1.65	4.04 ***	2.31–7.06	2.96 *	1.19–7.35
**Marital Status**						
Single	1.00		1.00		1.00	
Married	1.13	0.64–1.99	1.64	0.78–3.45	1.05	0.71–1.57
Cohabiting	-	-	-	-	-	-
Widowed	-	-	-	-	-	-
Divorced/separated	-	-	-	-	-	-
**Geopolitical zone**						
North central	1.00		1.00		1.00	
Northeast	3.16 ***	2.68–3.71	0.98	0.76–1.26	0.99	0.95–1.03
Northwest	1.31	0.94–1.82	0.83	0.67–1.03	0.59 **	0.44–0.79
Southeast	1.27	0.27–5.83	1.21 ***	1.12–1.31	0.95	0.54–1.69
South south	1.18	0.76–1.85	1.38 ***	1.28–1.46	1.10	0.70–1.71
Southwest	1.50	0.48–4.61	1.12	0.90–1.38	0.85	0.46–1.58
**Height**						
Below average	1.00		1.00		1.00	
Above average	1.07	0.98–1.16	0.89	0.76–1.04	0.03 ***	0.01–0.13
**Weight**						
Below average	−		−		1.00	
Above average	−	−	−	−	36,169.13 ***	7536.5–173,583.4
**Children borne**						
1–3	1.00		1.00		1.00	
4–6	0.77	0.68–0.88	1.03	0.89–1.18	1.41	0.88–2.25
7 and above	0.96	0.79–1.16	1.07	0.92–1.25	1.26	0.81–1.97
**Contraceptive method**						
None	1.00		1.00		1.00	
Folkloric	1.32	0.23–7.47	0.54 **	0.36–0.81	0.75	0.47–1.18
Traditional	0.73	0.23–2.27	1.79 ***	1.46–2.19	2.23 ***	1.79–2.76
Modern	0.94	0.41–2.11	1.61 **	1.21–2.14	1.27 *	1.03–1.57
**Breastfeeding**						
No	1.00		1.00		1.00	
Yes	1.27 ***	1.21–1.33	0.96	0.87–1.06	0.94	0.69–1.27
**Living with partner**						
No	1.00		1.00		1.00	
Yes	1.23	0.85–1.77	0.91 ***	0.89–0.94	1.05 ***	1.01–1.07
**Long working hours**						
No	1.00		1.00		1.00	
Yes	1.04	0.83–1.29	0.75 *	0.59–0.95	0.93	0.66–1.29
**Cigarette smoking**						
No	−	−	−	−	1.00	−
Yes	−	−	−	−	2.62	0.02–233.26
**Residence**						
Sd (cons)	0.21	0.20–0.21	0.31	0.31–0.32	0.62	0.61–0.62

Normal weight was the reference group, adjusted for all variables in the column (reference category = 1.00). * *p* < 0.05, ** *p* < 0.01, *** *p* < 0.001. OR (odd ratio); AOR (adjusted OR); UOR (unadjusted OR).

**Table 6 ijerph-19-00125-t006:** Multi-level analysis showing the predictors of body weight among women of childbearing age and factors by urban–rural variation in South Africa.

Factors	Underweight		Overweight		Obese	
	UOR	95% CI	UOR	95% CI	UOR	95% CI
**Age group**						
15–19	1.00		1.00		1.00	
20–24	0.032	0.001–10.2	0.32 **	0.14–0.70	51.46	0.61–113.42
25–29	0.051	0.002–12.2	0.45 **	0.25–0.79	43.84	0.58–104.32
30–34	0.010	0.001–2.31	0.61 *	0.37–0.99	22.72	0.31–91.17
35–39	0.00004 *	1.5 × 10^−8^–0.16	0.89	0.54–1.48	97.83	0.93–161.74
40–44	0.0036	7.1 × 10^−6^–1.82	1.05	0.65–1.67	63.47	0.83–117.48
45–49	0.0015 *	2.3 × 10^−6^–0.98	1.00	-	109.24	0.96–191.83
**Education**						
No education	1.00		1.00		1.00	
Primary	0.03	0.003–2.46	2.72 *	1.23–5.95	0.02 **	0.002–0.212
Secondary	1.79	0.04–71.96	6.64 ***	3.03–14.50	1.35	0.18–9.85
Tertiary	−	−	2.92 *	1.07–7.93	0.001 ***	0.0001–0.014
**Employment status**						
Unemployed	1.00		1.00		1.00	
Self–employed	0.44 **	0.28–0.70	1.38	0.82–2.33	11.98 **	1.86–77.09
Employed	−	−	2.44 *	1.17–5.03	161,820.6 ***	2560.6–1.2 × 10^7^
**Wealth index**						
Poorest	1.00		1.00		1.00	
Poorer	4.91	0.39–61.14	0.68	0.44–1.05	1.91	0.56–6.48
Average	8.82	0.45–171.9	1.41	0.91–2.16	0.20 *	0.04–0.89
Richer	208.16 **	6.2–6887.6	3.05 ***	1.83–5.07	0.56	0.09–3.29
Richest	−	−	0.61	0.34–1.09	0.02 **	0.001–0.35
**Marital status**						
Single	1.00		1.00		1.00	
Married	0.48	0.06–3.70	0.85	0.61–1.18	0.69	0.22–2.08
Cohabiting	-	-	-	-	-	-
Widowed	-	-	-	-	-	-
Divorced/Separated	-	-	-	-	-	-
**Provinces**						
Western Cape	1.00		1.00		1.00	
Eastern Cape	0.49	0.05–4.87	5.25 ***	2.36–11.64	144.32 ***	8.89–2342.66
Northern Cape	0.14	0.07–2.49	0.63	0.22–1.77	9.94	0.84–116.62
Free State	0.46	0.02–7.92	2.19	0.83–5.74	1477.11 ***	55.47–39,329.67
KwaZulu-Natal	1	−	4.87 ***	2.31–10.25	1717.30 ***	77.11–38,244.93
Northwest	0.12	0.005–2.74	3.05 **	1.35–6.89	1626.01 ***	109.41–24,163.2
Gauteng	1	−	1.73	0.90–3.33	14.51 *	1.41–148.97
Mpumalanga	0.04	0.002–0.69	2.63 *	1.23–5.60	58.84 **	3.77–916.98
Limpopo	0.74	0.03–15.36	4.56 ***	2.07–10.04	2.25	0.13–38.31
**Height**						
Below average	1.00		1.00		1.00	
Above average	0.10 *	0.018–0.66	0.39 ***	0.29–0.54	-	-
**Weight**						
Below average	1.00		1.00		1.00	
Above average	-	-	569.35 ***	99.6–3255.8	2.9 × 10^19^ ***	2.5 × 10^17^–3.4 × 10^21^
**Children borne**						
1–3	1.00		1.00		1.00	
4–6	0.09	0.004–2.24	0.51 ***	0.35–0.74	0.57	0.21–1.49
7+	273.08 **	6.2–11,859.8	1.28	0.65–2.50	0.0004 ***	1.2 × 10^−6^–0.0013
**Contraceptive method**						
None	1.00		1.00		1.00	
Traditional	1	-	1	-	1	-
Modern	0.49	0.06–3.66	1.43 *	1.05–1.92	0.66	0.22–1.98
**Breastfeeding**						
No	1.00		1.00		1.00	
Yes	0.82	0.09–6.73	1.69 *	1.07–2.63	22.68 **	3.48–147.47
**Currently living with partner**						
No	1.00		1.00		1.00	
Yes	17.75	0.67–464.7	1.81 **	1.22–2.67	0.93	0.32–3.70
**Long working hours**						
No	1.00		1.00		1.00	
Yes	5.44	0.51–57.14	0.82	0.54–1.23	0.29 *	0.08–0.96
**Cigarette smoking**						
No	1.00		1.00		1.00	
Yes	2.14 **	1.27–3.61	0.41 **	0.21–0.78	145.99 ***	16.22–1313.38
**Residence**						
Sd (cons)	0.16	0.16–0.17	0.29	0.28–0.29	0.48	0.48–0.49

Normal weight was the reference group, adjusted for all variables in the column (reference category = 1.00). * *p* < 0.05, ** *p* < 0.01, *** *p* < 0.001. OR (odd ratio); AOR (adjusted OR); UOR (unadjusted OR).

**Table 7 ijerph-19-00125-t007:** ICC for body weight among women of childbearing age in Nigeria and South Africa.

Multilevel Model for Nigeria BMI Categories ‘ICC Estimate’			
Models	ICC	Standard Error	95% CI
Model 1 (Underweight)	0.0127	0.0003	0.0121–0.0133
Model 2 (Overweight)	0.0289	0.0001	0.0288–0.0289
Model 3 (Obese)	0.1040	0.0001	0.1038–0.1041
**Multilevel Model for South Africa BMI Categories ‘ICC Estimate’**			
**Models**	**ICC**	**Standard Error**	**95% CI**
Model 1 (Underweight)	0.0102	0.0002	0.0092–0.0121
Model 2 (Overweight)	0.0271	0.0001	0.0238–0.0316
Model 3 (Obese)	0.0819	0.0007	0.0801–0.0864

Source: Computation from the NDHS 2018 and 2016 SADHS.

## Data Availability

Data are from the Demographic and Health Survey and the dataset is open to qualified researchers free of charge. In order to access the data from DHS Measure, a written request was submitted to the DHS MACRO and permission was granted to use the data for this survey. To request access to the dataset, please apply at https://dhsprogram.com/data/dataset_admin/login_main.cfm?.

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
