# Peer review of "Multilevel Analysis of Urban–Rural Variations of Body Weights and Individual-Level Factors among Women of Childbearing Age in Nigeria and South Africa: A Cross-Sectional Survey"

_ijerph, 2021, doi:10.3390/ijerph19010125_

Round 1

Reviewer 1 Report

The description of the statistical analysis is too detailed. You can delete some sentences like: „… Odds Ratio greater than 1.00 implied that the var-iable increases the likelihood of the outcome (body weight) while it is the opposite when the OR is less than 1.00.”,  and “Theoretically, as S2w (within-cluster var-iance) moves towards 0 (ze-ro), ICC gets closer to 1 (one)”.

Figure 1-3 may be omitted as nearly all information is described in the manuscript.

The following text should be supported with OR values: “Also, employment status was found to be associated with overweight and obesity among women in Nigeria.[….] However, in urban and rural areas, women who smoked cigarettes had higher odds of being obese.”

Table 2, 2b, 3a, 3b - mark the reference categories, e.g. (base group).

Table 3a, 3b - explain the UOR abbreviation under the table.

Author Response

Review Report Form

Author's Reply to the Review Report (Reviewer 1) - Comments and Suggestions for Authors

Dear Reviewer 1,

             Thank you for all your efforts in reading my entire manuscript and the useful comments and suggestions you have made on our submitted manuscript. These suggestions are very important and has improve the content of the entire manuscript. However, the responses to the comments and suggestions were made directly in the main manuscript and it is made in RED COLOUR in the main manuscript.

  1. Reviewer 1 Comment(s): The description of the statistical analysis is too detailed. You can delete some sentences like: „… Odds Ratio greater than 1.00 implied that the variable increases the likelihood of the outcome (body weight) while it is the opposite when the OR is less than 1.00.”, and “Theoretically, as S2w (within-cluster variance) moves towards 0 (zero), ICC gets closer to 1 (one)”.

Author(s) Responses: [Odds Ratio greater than 1.00 implied that the variable increases the likelihood of the outcome (body weight) while it is the opposite when the OR is less than 1.00.”,] – Please, this sentence cannot be deleted as this is the key interpretation of the analysis/findings of the Multivariate model. Without this sentence, it is difficult for one who does not have a prior knowledge of the Multivariate model/analysis will not be able to understand the interpretation of the multivariate model.

[“Theoretically, as S2w (within-cluster variance) moves towards 0 (zero), ICC gets closer to 1 (one)”] – Also, this sentence cannot be deleted as this is the key interpretation of the analysis/findings of the ICC. Without this sentence, it is difficult for one to understand the table involving ICC without the sentence above.

  1. Reviewer 1 Comment(s): Figure 1-3 may be omitted as nearly all information is described in the manuscript.

Author(s) Responses: Figure 1 cannot be omitted as Figure 1 illustrates the prevalence or the widespread or the widespread or the epidemiology of body weights. Thus, body weight has been a public health concern globally, and in some parts of sub-Saharan African countries such as in South Africa and Nigeria. To improve this growing problem, it is important to understand the prevalence in contemporary body weights in South Africa and Nigeria. Therefore, before we can attain such goals, it is necessary to elucidate the current prevalence of body weights across diverse populations. One of the objectives of the study is to determine the current national prevalence of body weights in both countries, Along with how they are changing overtime.

                                                                           Normal weight        Underweight      Overweight       Obese

                                     South Africa                     25.1%                       1.8%                  28.2%               44.9%

                                     Nigeria                              59.4%                       9.0%                  20.2%               11.4%

Figure 2 and Figure 3 cannot be omitted as both figures illustrates the prevalence of body weights among women of childbearing age by urban-rural variations in Nigeria (Figure 2) and the prevalence of body weights among women of childbearing age by urban-rural variations in South Africa (Figure 3). This is important as body weight differences between urban-rural variations should be reported in Nigeria (Figure 2) and in South Africa (Figure 2). It is unknown whether these are due to effects of social selection or social causation are not found in the 2018 NDHS and 2016 SADHS. Hence, the prevalence findings are very key for policy implications for both countries.

  1. Reviewer 1 Comment(s): The following text should be supported with OR values: “Also, employment status was found to be associated with overweight and obesity among women in Nigeria. [….] However, in urban and rural areas, women who smoked cigarettes had higher odds of being obese.”

Author(s) Responses:

[The following text should be supported with OR values: “Also, employment status was found to be associated with overweight and obesity among women in Nigeria]. The text was added with OR values for the employment status:

Also, for employment status in Nigeria, self-employed respondents were found to be associated with being underweight (OR = 1.49; p<0.001), while self-employed (OR = 4.70, p<0.01) and employed (OR = 9.02; p<0.001) respondents were found to be associated with obesity, respectively.

[However, in urban and rural areas, women who smoked cigarettes had higher odds of being obese.”] This is applied also. See below:

However, in urban-rural areas, women who smoked cigarettes had higher odds of being obese (OR = 4.52; p<0.001).

  1. Reviewer 1 Comment(s): Table 2, 2b, 3a, 3b - mark the reference categories, e.g. (base group).

Author(s) Responses: In the manuscript, we have Table 1, Table 2a, Table 2b, Table 3a, Table 3b and Table 4. Therefore, the reference categories is marked as 1.00 in Table 2a, Table 2b, Table 3a and Table 3b. For instance, ‘Age group’, 15–19 is the reference category marked 1.00 under AOR in the table.

  1. Reviewer 1 Comment(s): Table 3a, 3b - explain the UOR abbreviation under the table.

Author(s) Responses: In table 3a and 3b, UOR (unadjusted OR) abbreviation has been explained under the table.

Reviewer 2 Report

The authors analyze the relationship between body mass index and several environmental and individual covariates, considering survey data from Nigeria and South Africa. The authors obtain useful results, but I am unsure about the scientific soundness of the study. The authors should emphasize this. Besides, I have the following concerns, related to formatting issues mainly:

  • I think Section 2.4 could be replaced by a summary table.
  • The Equations provided in Section 2.5 need to be rewritten in a better format (use Microsoft Word’s Equation Editor). Now, it is difficult to understand the modeling approach for this reason. Besides, the authors should be more careful when describing the model and use ln (P/(1-P)) (with parenthesis) to specify the logit link.
  • The text still needs a revision in terms of grammar and style. It would be better to shorten some sections, as the paper is quite dense at the moment.

Author Response

Review Report Form

Author's Reply to the Review Report (Reviewer 2) - Comments and Suggestions for Authors

Dear Reviewer 2,

          Thank you for taking out time to read our entire manuscript. We acknowledge all your comments and suggestions you have made on our submitted manuscript. Your comments and suggestions are very important and after revising, they have improve the content of the manuscript.  

           However, the responses to the comments and suggestions were made directly in the main manuscript and it is made in BLUE COLOUR in the main manuscript.

  1. Reviewer 2 Comment(s): The authors analyze the relationship between body mass index and several environmental and individual covariates, considering survey data from Nigeria and South Africa. The authors obtain useful results, but I am unsure about the scientific soundness of the study.

Author(s) Responses: To the best of the Authors’ knowledge, this study has showed the scientific soundness of this study through its validity (the capacity of the variables measured – which are the outcome and the explanatory variables), reliability of this study in its capacity in providing established findings/outcomes for study replication in other geographical locations, and explicitness of the scientific base/evidence that supports the use of variables and data gotten from national representative demographic health surveys for Nigeria and South Africa. The outcome and explanatory variables and their cutoffs for the body weights were based on international indicators according to the World Health Organization (WHO) recommendation; and also they have been previously used in demographic health surveys as well as other studies conducted across African countries. This study has provided useful insights into the importance of using demographic health survey data and findings from the scientific study, and this is sufficient in establishing the relative merit and scientific soundness of this study. In addition, establishing a comparison of body weights between two countries (Nigeria and South Africa) was demonstrated in clarity in terms of thought processes during the data analysis and subsequent interpretations of the study analysis since 2018 NDHS and 2016 SADHS was engaged for both countries which helps in reducing research bias. This study seeks out the similarities and differences in bodyweights between both countries that exists across different interpretations to ensure diverse perspectives, which are represented including in-depth description and justifications in the methodology and analysis to support the present findings of this study.

The authors should emphasize this. Besides, I have the following concerns, related to formatting issues mainly:

  • Reviewer 2 Comment(s): I think Section 2.4 could be replaced by a summary table.

    Author(s) Responses: Although, narrative of the measurement of the independent variables would have been better but we just follow your suggestions, and we draw a table to show the summary and it has been inserted in the main manuscript as you have suggested.

  • Reviewer 2 Comment(s): The Equations provided in Section 2.5 need to be rewritten in a better format (use Microsoft Word’s Equation Editor). Now, it is difficult to understand the modeling approach for this reason. Besides, the authors should be more careful when describing the model and use ln (P/ (1-P)) (with parenthesis) to specify the logit link.

    Author(s) Responses: The Equations have been rewritten in a better format using Microsoft Word’s Equation Editor. That is it below and it has been inserted in the main manuscript.

The modeling approach has been described in a clear way for better understanding. See that above.

  • Reviewer 2 Comment(s): The text still needs a revision in terms of grammar and style. It would be better to shorten some sections, as the paper is quite dense at the moment.

    Author(s) Responses:  The content of the text was edited by a Professional Language Editor. However, if you say my text still needs a revision in terms of grammar and style, I have made an arrangement with MDPI English language Editor for thorough editing of the grammar and style.

Reviewer 3 Report

Multi-level analysis of urban-rural variations of body weights and
individual-level factors among women of childbearing age in Nigeria and South Africa: a cross-sectional survey

ABSTRACT:

-Don´t use abbreviations in the abstract

-Attention to breaking down words at the end of the sentences, i.e., cig-arette, indi-vidual, etc.

-The conclusion is not clear enough and it´s not connected to the results, rewrite it please. There is also some confusion in the keywords in the last paragraph, please correct it.

INTRODUCTION

-Attention to breaking down words at the end of the sentences, i.e., un-derweight, go-ing, stud-ies, preva-lence, coun-tries, repre-sentative, na-ture, pop-ulation,  etc.

-Introduce abbreviations the first time they appear in the text, i.e., NDHS

-Delete the explanation the second time you use the abbreviation ICC

MATERIAL AND METHODS

-Attention to breaking down words at the end of the sentences, i.e., reli-gions, wom-en, col-lected, rec-orded, categ-orical, char.acteristics, vari-ables, var-iance, etc.

-Explain Boko Haram conflicts or delete it

-Delete the explanation the second time you use the abbreviation IBM

-Introduce abbreviations the first time they appear in the text, i.e., SADHS

Study design: third line is unclear “Besides increasing the number of observations, another advantage…”

-Merging to databases from two different geographical areas might include bias, i.e., merging two different populations. A generalized linear mixed model can address the differences that might arise from different countries. Please, explain in detail this approach.

-2.3.2.-Independent variables: references are not correctly inserted

RESULTS

-Explain why you didn´t use age and body weight as a continuous variable.

-Regarding employment status, more than a half of Nigerian women were self-employed (92.7%)…this percentage should read “Almost all Nigerian women were…”

-Table 1, Table 2, Table 3 and Table 4: be consistent ad use just one decimal in numbers in the same way you have done in the manuscript.

-Use the commas in the numbers when appropriate (i.e., table 1) not only as total numbers

DISCUSSION

-line 165, pag 8 Use an uniform reference style, etc. All the discussion section has another bibliographical system compared to the introduction section.

Author Response

Review Report (Reviewer 3)

Author's Reply to the Review Report (Reviewer 3) - Comments and Suggestions for Authors

Multi-level analysis of urban-rural variations of body weights and
individual-level factors among women of childbearing age in Nigeria and South Africa: a cross-sectional survey

Dear Reviewer 3,

Thank you for reviewing our submitted manuscript to IJERPH. We acknowledge all your useful comments and suggestions you have made on our submitted manuscript. These suggestions are very vital and has been effected in the main manuscript. This has further improve the content of the manuscript.

 However, the responses to the comments and suggestions were made directly in the main manuscript and it is done in GREEN COLOUR in the main manuscript.

ABSTRACT:

  1. Reviewer 2 Comment(s): -Don´t use abbreviations in the abstract

Author(s) Responses: This is noted and it has been effected in the abstract.

  1. Reviewer 2 Comment(s): -Attention to breaking down words at the end of the sentences, i.e., cig-arette, indi-vidual, etc.

Author(s) Responses: This is noted and it has been effected in the entire manuscript. I think this could be an issue from the journal who had an older version of the Microsoft word.

  1. Reviewer 2 Comment(s): -The conclusion is not clear enough and it´s not connected to the results, rewrite it please. There is also some confusion in the keywords in the last paragraph, please correct it.

Author(s) Responses: This is noted and the conclusion has been revised in order for it to be clear.

INTRODUCTION

  1. Reviewer 2 Comment(s): -Attention to breaking down words at the end of the sentences, i.e., un-derweight, go-ing, stud-ies, preva-lence, coun-tries, repre-sentative, na-ture, pop-ulation, etc.

Author(s) Responses: This is noted and it has been effected in the entire manuscript. I think this could be an issue from the journal who had an older version of the Microsoft word.

  1. Reviewer 2 Comment(s): -Introduce abbreviations the first time they appear in the text, i.e., NDHS

Author(s) Responses: This is noted and it has been effected in the entire manuscript. All abbreviations have been given in full first, then abbreviations were introduce much later in the manuscript. NDHS has been spelt in full, subsequently, abbreviations start to appear

  1. Reviewer 2 Comment(s): -Delete the explanation the second time you use the abbreviation ICC

Author(s) Responses: This is noted but the ICC explanation was done once in the methods section. ICC was introduced the first time but the detailed explanation for ICC was done second section.

MATERIAL AND METHODS

  1. Reviewer 2 Comment(s): -Attention to breaking down words at the end of the sentences, i.e., reli-gions, wom-en, col-lected, rec-orded, categ-orical, char.acteristics, vari-ables, var-iance, etc.

Author(s) Responses: This is noted and it has been effected in the entire manuscript. I think this could be an issue from the journal who had an older version of the Microsoft word.

  1. Reviewer 2 Comment(s): -Explain Boko Haram conflicts or delete it

Author(s) Responses: Boko Haram is one of the Islamist militant groups in Nigeria that has carried out several terrorists’ attacks on religious and political groups, local police, and the military, as well as indiscriminately attacking civilians in busy markets and villages. Presently, Nigeria is plagued with Boko Haram conflicts, poverty, malnutrition, and dis-eases and the burden of youth unemployment and it is important to state it as part of Nigeria demography. It cannot be deleted.

  1. Reviewer 2 Comment(s): -Delete the explanation the second time you use the abbreviation IBM

Author(s) Responses: This is noted and it has been effected.

  1. Reviewer 2 Comment(s): -Introduce abbreviations the first time they appear in the text, i.e., SADHS

Author(s) Responses: This is noted and it has been effected.

  1. Reviewer 2 Comment(s): Study design: third line is unclear “Besides increasing the number of observations, another advantage…”

Author(s) Responses: This is noted and it has been effected.

  1. Reviewer 2 Comment(s): -Merging to databases from two different geographical areas might include bias, i.e., merging two different populations. A generalized linear mixed model can address the differences that might arise from different countries. Please, explain in detail this approach.

Author(s) Responses: This approach means that generalized linear mixed models are an extension of linear mixed models to allow response variables from different distributions, such as binary responses. The assumptions of the generalized linear mixed model involves validity of the model, independence of the data points, linearity of the relationship between predictor and response, absence of measurement error in the predictor, homogeneity of the residuals, independence of the random effects versus covariates (erogeneity).

  1. Reviewer 2 Comment(s): -2.3.2.-Independent variables: references are not correctly inserted

Author(s) Responses: This is noted and it has been effected as [25-29].

RESULTS

  1. Reviewer 2 Comment(s): -Explain why you didn´t use age and body weight as a continuous variable.

Author(s) Responses: The reason why age and body weight is not used as a continuous variable in this study is that: First, the body weight derived from the body mass index (BMI) was the dependent variable. It was calculated by dividing the body weight in kilograms by height squared (m2). BMI was categorized into four categories, namely: Underweight (BMI < 18 kg/m2); Normal weight (18kg/m2 ≤ BMI < 25 kg/m2); Overweight (25 kg/m2 ≤BMI < 30 kg/m2); and Obese (BMI ≥ 30 kg/m2) according to the World Health Organization’s (WHO) recommendation. Therefore, body weight as a continuous variable cannot be feasible. Second, age as a continuous variable will not make a meaningful analysis as it will not bring a presentable results like categorical variables. Third, if age is treated as continuous, some age will be missing or too small in numbers. Third, we have multiple. Fourth, the analysis to be used is based on the outcome variable and age as a continuous variable will not fit into logistic regression model. Thus, categorical variable has values that can be put into a countable number of distinct groups based on a characteristic. Logistic regression transforms the dependent variable and then uses Maximum Likelihood Estimation, rather than least squares, to estimate the parameters by describing the relationship between a set of independent variables and a categorical dependent variable.

  1. Reviewer 2 Comment(s): -Regarding employment status, more than a half of Nigerian women were self-employed (92.7%)…this percentage should read “Almost all Nigerian women were…”

Author(s) Responses: Regarding employment status, almost Nigerian women were self (92.7%) compared to unemployed women (75.3%) in South Africa.

  1. Reviewer 2 Comment(s): -Table 1, Table 2, Table 3 and Table 4: be consistent ad use just one decimal in numbers in the same way you have done in the manuscript.

Author(s) Responses: Corrections have been made on the tables as follows such as Table 1, Table 2a and Table 2b, Table 3a and Table 3b and Table 4. The description of tables 1 to 4 are as follows:

Table 1. Socio-demographic, geographical and behavioural characteristics of women in Nigeria and South Africa.

Table 2a. Bivariate analysis showing adjusted odds of ratio of body weights and associated factors among women of childbearing age by urban-rural variations in Nigeria.

Table 2b. Bivariate analysis showing adjusted odds ratio of body weight and its associated factors among women of childbearing age by urban-rural variations in South Africa.

Table 3a. Multi-level analysis showing the predictors of body weight among women of childbearing age and factors by urban-rural variation in Nigeria.

Table 3b. Multi-level analysis showing the predictors of body weight among women of childbearing age and factors by urban-variation in South Africa.

Table 4. ICC for body weights among women of childbearing age in Nigeria and South Africa.

  1. Reviewer 2 Comment(s): -Use the commas in the numbers when appropriate (i.e., table 1) not only as total numbers

Author(s) Responses: This has been effected in Table 1.

DISCUSSION

  1. Reviewer 2 Comment(s): -line 165, page 8 Use an uniform reference style, etc. All the discussion section has another bibliographical system compared to the introduction section.

Author(s) Responses: The reference style has been corrected and it is uniform in the entire manuscript.

Round 2

Reviewer 2 Report

The authors have improved the manuscript in terms of presentation and grammar, but the format of the Equations is still unsuitable. Please use subindices for the variables and coefficients and a better style. I don't have any other comment.

Author Response

Comments and Suggestions for Authors – Reviewer 2

Dear Reviewer 2, Thank you for your comments and all these comments/suggestions have improved our manuscript better. However, the responses to the comments/suggestions were made in BROWN COLOUR and they are as follows:

Reviewer 2 Comments: The authors have improved the manuscript in terms of presentation and grammar, but the format of the Equations is still unsuitable. Please use sub-indices for the variables and coefficients and a better style. I don't have any other comment.

Author(s) Response: I (the Corresponding Author) use the Equation tools to create the equation. However, there is no separate Subscript or Superscript tool in the equation tools, except the ones that comes with an already equation. When using the Equation platform, the Subscript or Superscript tools in ‘HOME’ in Microsoft Word will not work. And even if I did it on Microsoft Word, if I copy it to paste on the Equation space box, it will not copy.

Also, the typesetting of the paper also made the equation to be long, but I have tried to adjust it. I know that by the time the IJERPH Production Editor/Publisher will start copyediting and typesetting, they will assist in the Equation and they will use a better software to use sub-indices for the variables and coefficients and that will improve the equation to make it a better style as you have suggested.

However, I have made some adjustments to the Equation, I pray and hope it makes a better style. See below:
